# Microstructural Characterization and Mechanical Properties of Laser Beam-Welded Dissimilar Joints between A6000 Aluminum Alloy and Galvanized Steel

**DOI:** 10.3390/ma15020543

**Published:** 2022-01-12

**Authors:** Nkopane Angelina Ramaphoko, Samuel Skhosane, Nthabiseng Maledi

**Affiliations:** 1School of Chemical and Metallurgical Engineering, University of the Witwatersrand, Private Bag 3, Johannesburg 2050, South Africa; Nthabiseng.Maledi@wits.ac.za; 2Laser Enabled Manufacturing National Laser Centre CSIR, Pretoria 0001, South Africa; SSkhosane@csir.co.za

**Keywords:** laser welding, heat input, IMCs, microstructure, mechanical properties

## Abstract

This paper presents the laser beam welding process of a lap joint between galvanized steel (Z225) and an aluminum alloy (A6000) from an IPG fiber laser. Welding of steel to aluminum has become popular in the automotive industry as a means of reducing the total vehicle body mass. This approach reduces fuel consumption and, ultimately, carbon emissions. Laser welding parameters used to control heat input for the study were laser power ranging between 800 and 1200 W, as well as laser welding speeds between 2 and 4 m/min. Distinct features of the dissimilar joints were microscopically examined. The SEM-EDS technique was employed to study the intermetallic phases along the Fe-Al interface. The outcome revealed the presence of “needle-like phases” and “island-shaped phases” at high heat inputs. Traces of both Fe_2_Al_5_ and FeAl_3_ phases were detected. For low heat input, there was evidence of insufficient fusion. Weld width was influenced by welding parameters and increased with an increase in heat input. Mechanical properties of the joints indicated that the microhardness values of the weld joints were higher than those of both base metals. The maximum tensile shear strength obtained was 1.79 kN for a sample produced at 1200 W and 3 m/min.

## 1. Introduction

In the past, car manufacturers produced heavy vehicles in order to meet passenger safety and aesthetic requirements [1]. The cars were heavy due to the predominant usage of steel for most car body parts. There has been an introduction of more stringent rules to protect the environment by lessening the amount of car exhaust emissions such as CO_2_ and NO_x_. Therefore, the automobile industry strives to decrease the total mass of vehicles as there is strong evidence that this measure is one of the most effective methods for reducing fuel consumption and consequently decreasing the release of harmful greenhouse gases. The New European Driving Cycle (NEDC) reported that there is a proportional relationship between vehicle weight and fuel consumption. It was stated that decreasing vehicle weight by 100 kg could save fuel by 0.3–0.5 L/100 km [2].

Initially, the approach used to address the matter was to reduce the thickness of the steel sheets used for car panels. However, the method was unsuccessful as it decreased the strength of the cars and thus compromised passenger safety. In recent times, it has been discovered that the most effective means of decreasing the car mass is by replacing some steel parts with lighter materials such as aluminum alloys [2]. As a result, the application of the method within the car manufacturing sector requires the joining of dissimilar materials. Aluminum alloys that are mostly used for this purpose are the 5000 and 6000 series alloys because they possess exceptional corrosion properties, suitable strength, and excellent surface characteristics [3]. Welding of dissimilar materials is a process of joining materials with different physical properties by means of a heat source [4]. The technique allows for the manufacturing of weld joints with required traits for certain applications. Nevertheless, welding of dissimilar materials presents difficulties due to the formation of inevitable hard and brittle intermetallic compounds (IMCs) at the weld interface, which adversely impacts the soundness of the weld joint [5].

Research has verified the success of laser welding of dissimilar materials over conventional welding processes because it is characterized by thin heat-affected zone (HAZ), high energy density, less deformation, high welding speed, and great control of heat input [6]. Furthermore, the quality of dissimilar metals weld joints may be enhanced by a suitable selection of laser welding parameters. Sierra et al. [7] emphasized the significance of laser welding parameters from the conclusions of their research. It was discovered that various arrangements of welding parameters influence the IMC layer thickness. It was concluded that increasing laser welding speed reduced the IMC layer thickness. Tilekar et al. [8] also conducted similar research; however, the focus was on the analysis of the weld bead dimensions (weld penetration depth and weld width). It was found that the welding speed and laser power are the parameters that have a significant impact on the weld width and weld penetration [8]. In research by Torkamany et al. [9], a pulsed Nd: YAG laser was applied to combine a 0.8 mm-thick steel to a 2.0 mm-thick aluminum alloy [9]. The impact of laser power and pulse duration on the soundness of the weld joint was examined. The results revealed the presence of IMCs within the weld zone. It was stated that the mechanical properties of dissimilar weld joints are substantially influenced by the composition of parent metals, welding parameters, and shape contacts of the weld bead, which are impacted by input welding parameters [10].

Several efforts have been made to optimize the welding parameters in order to attain weldments of high quality, acceptable geometry, and low surface roughness. Cheohlee et al. [11] noticed that the tensile strength is mostly influenced by welding speed, followed by laser power and the focal point. It was also concluded that the parameters that produced appropriate microstructure and mechanical properties were laser power ranging between 0.6 and 2.4 kW, welding speed ranging between 0.82 and 3 m/min, focal position between −0.8 and −0.2 mm, and shielding gas of flowrate between 5 and 15 L/min [11].

The aim of the experiment by Zhang et al. [12] was to achieve deep penetration from a high-intensity fiber laser of 10 kW intensity. The process parameters under the spotlight were welding speed, focus lens, focus size, shielding gas, and laser power. Shielding gases helium, nitrogen, and argon (He, N_2,_ and Ar) were considered as output parameters [12]. SEM analysis and a transverse tensile strength test were also conducted. The findings revealed that it was difficult to achieve full weld penetration by proper focal position when thick samples are used. Helium gas resulted in the greatest penetration, followed by N_2_ and Ar. This is because helium has the lowest density and has better ionization energy compared to the other two gases [12].

One of the major challenges encountered during laser fusion welding between galvanized steel and aluminum alloys is the creation of pores within the weld seam, especially when using the keyhole technique [13]. This is caused by the fact that the melting and boiling points of Zn are lower than those of Al and Fe. Since high temperatures are applied during fusion welding, the zinc layer covering the steel tends to vaporize and become trapped during solidification creating pores [13,14]. In order to control the escape of Zn vapor, a special clamping system that has a tiny opening at the aluminum-steel interface may be used to allow the Zn vapor to escape and thus reduce pore formation [15]. However, the advantage of using galvanized steel in reactive welding methods is that the zinc layer improves the wettability of molten aluminum on the solid steel [13]. It has also been shown that the shear strength of the weld joint can be enhanced to values as high as 200 MPa by regulating the laser welding parameters such as the heat source location to control the exclusive melting of base metals.

Numerous articles on different combinations of dissimilar materials for application within the automotive industry have been reported. However, little work has been reported on the combination of Z225 and A6000 specifically. As a result, the purpose of the current work was to determine the optimum laser welding parameters for producing Z225 and A6000 joints with microstructural and mechanical properties suitable for manufacturing vehicle body parts.

## 2. Materials and Methods

### 2.1. Materials and Laser Welding Method

Galvanized steel (Z225) and an aluminum alloy (A6000) were used as parent/base materials. The zinc coating on the steel plate was 60 g/cm^2^. The samples were cut using a CO_2_ laser cutter into plates of 50 mm length and 30 mm width. The thicknesses of Z225 and A6000 plates were 0.8 and 1.2 mm, respectively. The elemental composition and mechanical characteristics of the parent metals are given in Table 1 and Table 2, respectively.

Acetone was used to remove the impurities from the surfaces of the plates, as they tend to melt during welding and create defects. The plates were then arranged into a lap joint with a 3 mm-wide overlap. A toggle system on a fixed gantry table was used in order to maintain constant pressure during welding and to prevent a microscopic gap between the metal plates for maximum heat conduction. A Precitec YW 50 welding head (made by Precitec Engineering in Lahore, Pakistan) was mounted to a KUKA KR60L30HA automated arm robot manufactured by KUKA located in Gersthofen, Germany to perform the laser welding process. The weld seam was created from an IPG fiber laser with a 1.06 µm wavelength and 3000 W maximum power at a −5 mm defocused laser beam and 171 mm focusing lens. An inert helium gas discharged at a 15 L/min flowrate was used as a shielding gas to avoid oxidation on the heated surfaces. A 12 mm gap between the laser head and workpiece was maintained throughout the joining process.

The laser-welded dissimilar lap joints were exposed to destructive testing methods for characterization. A Struers Secotom-10 abrasive cutting machine (manufactured by Struers in Ballerup, Denmark) was used to cut the cross-section of the weld joints for microscopic and macroscopic evaluation. The machine was set at a feed speed of 120 mm/s and 3500 r/min wheel speed. Black phenolic resin was used to mount the samples in a hot mounting press for 5 min at 200 °C. The samples were ground and polished with 220, 800, and 1200 SiC, respectively using a Saphir 520 single wheel grinder/polisher with Rubin 500 head (by Unitron Instrumentation Technology Pvt. Ltd in Bengaluru, India) in order to achieve a mirror finish surface. The surfaces were then etched with 3% Nital solution (3% HNO_3_, 97% ethyl alcohol) to expose the microstructure and break down the grain boundaries.

### 2.2. Microstructure

The weld microstructural analysis was performed using an Olympus SC 50 optical microscope by Olympus Life Science in Waltham, MA, USA at magnifications between 10× and 50× to examine the grain sizes. The weld bead dimensions were measured from the installed AxioVision imaging software. Three measurements were taken on the same area, and the average value was used. The samples were then cut to less than 5 mm sections. Scanning electron microscope (SEM) with electron dispersive spectrum (EDS) (Carl Zeiss MGA Sigma FESEM) produced by Carl Zeiss Microscopy in White Plains, NY, USA was then employed. The machine was set at 20 kV electron high tension (EHT). The SEM was used to examine the morphology and composition of the dilution zone. The EDS was then used to detect the present elements at different sections. A Bruker D8 advanced X-ray diffractometer (Manufactured by Bruker in Bremen, Germany) with a monochromator was used to perform an X-ray Diffractor (XRD) analysis. The IMC phases were identified from the spotted elemental peak using the Diffract Eva software (Version 5.2.0.3). The equipment used a monochromator with a solid detector as well as a Cu Kα radiation source with a wavelength of 1.5406, step size of 0.02° and theta ranging between 2.5° and 45°. The voltage and current applied were 40 kV and 40 mA, respectively.

### 2.3. Mechanical Properties

Mechanical tests were performed on the samples to determine the impact of welding parameters.

#### 2.3.1. Microhardness Testing

A microhardness tester FM-700 (by Vickers in Shanghai, China at 300 gf load and 10 s dwell time was used to establish hardness profile along a horizontal line crossing from the Z225 base metal through the weld joint toward the A6000 base metal. A total of 16 indentations were performed on the Z225 base metal, weld metal, and the A6000 base metal for each sample.

#### 2.3.2. Tensile Strength Testing

Samples were cut into specimens of 0.9 mm width and 75 mm length to conduct a tensile shear strength test, which was conducted in accordance with ASTM E8 geometry. A Zwick/Roell universal testing machine (manufactured by Zwick/Roell Group in Ulm, Germany) was used to determine the strength of weld joints. The experiment was carried out at 25 °C temperature, 1 mm/s crosshead speed, and a nominal force of 20 kN load. The direction of the applied stress was perpendicular to the weld seam, as shown in Figure 1.

### 2.4. Experimental Principle

Laser welding was performed during the experiments. A lap joint was arranged by positioning steel on top of aluminum. The flow of heat infiltrates through the steel plate toward the aluminum by means of a positive gradient of thermal conductivity [17]. The melting point of aluminum is significantly lower than that of steel, and this makes it possible to liquefy aluminum, whereas steel is heated but retains its solid form. The power density of the laser spot subjected to the work piece determines the mode in which the laser beam gets into contact with the material [18]. The schematic representation of the experimental hypothesis in Figure 2 demonstrates a typical lap joint produced by conduction mode. In order to regard a weld in conduction mode, the power density should not be greater than 10^6^ W/cm^2,^ and the aspect ratio should be smaller than 0.5 [18]. No filler material was added in all the experiments.

### 2.5. System Parameters

The aim of the investigation was to determine the effect of laser welding parameters on the quality of a dissimilar joint between Z225 and A6000. Heat input was controlled by altering laser welding speed and laser power. Equation (1) was used to calculate the heat input values. Increasing laser power and decreasing laser welding speed increases heat input. The combinations of parameters used are recorded in Table 3. A constant beam diameter of 13 mm was used for producing all the samples [19].
Heat input (J/mm) = Laser power (W)/Welding speed (m/min)(1)

## 3. Results and Discussion

### 3.1. Microstructure

The optical micrographs showing the cross-sections of the weld joints produced under different conditions are depicted in Figure 3 and Figure 4. Figure 3b revealed evidence of undercutting. The defect most likely resulted from the weld metal failing to sufficiently wet the base metals due to either incorrect cleaning of the base metals or beam asymmetry [20]. In some cases, undercut defect is caused by evaporation of the zinc coating as heat input increases. The Zn vapor tends to accumulate pressure, which induces disturbances on the flow of the melt, causing the undercut [8]. In Figure 4b,c, it can be noticed that there was insufficient fusion due to a reduction in heat input. Pores were also detected for weld joint in Figure 4b. This may have been due to evaporation of the Zn-coating and entrapment of the vapor during solidification. It can be observed that the grains for the weld metal produced at higher heat inputs (Figure 3c and Figure 4a) have a coarse columnar structure. This was caused by grain growth due to high heat input, which typically results in a deficit of ductility [21].

Furthermore, irregular and complex phases were detected for weld joint produced at 1200 W and 3 m/min. These phases were analyzed at higher magnification (see Figure 5b). Island-shaped structures were observed at the A6000/weld interface. These resulted from the occurrence of conduction at the weld pool because of higher heat input [22]. The increased mobility of the molten pool increased the contact of steel and aluminum atoms and increased the entrapment of Al and Zn. In addition, between the “islands” and the weld metal, needle-like phases were detected within the Al substrate. Similar results were observed by Cui et al. [18]. They detected complex and irregular structures with several needle-like shapes within the microstructure near the steel/Al interface. The creation of the structures was attributed to high welding heat input [18]. The frequency of formation of such irregular phases generally increases with an increase in weld penetration [23].

The influence of laser power on the microstructure of the weld metal was further analyzed and observed at higher magnification, as displayed in Figure 6. It can be observed that the grain structure for both 800 and 1000 W was similar. Both samples are characterized by a fine microstructure. As power was increased to 1200 W, the grain size increased, and the structure changed from a fine to a coarser equiaxed microstructure in the middle of the weld metal. Columnar grains were observed at the weld and base metal interface. Epitaxial grain growth was evident within the weld metal, and the grain orientation was random. The observed differences in microstructure of the weld metal in the middle and the edges are often caused by the fact that in laser beam welding, the melt undergoes a rapid transition from liquid to solid in a short period. This causes different regions of the weld to undergo different solidification rates, which create variations in microstructure [24].

Figure 7 displays the effect of laser welding speed on the grain structure of the fusion zone.

The size of the grains at 2 m/min was large and decreased as the laser welding speed increased to 4 m/min. When laser welding speed was 3 and 4 m/min, a fine columnar microstructure was observed. The microstructure of the weld was determined by welding speed as it influences the weld pool size and shape [25]. As the weld pool becomes larger, the grain structure becomes coarser. In general, the grain size of the weld metal decreases with an increase in laser welding speed [25]. This is because low laser welding speed leads to high heat input, which results in slow cooling rates. Therefore, grain growth occurs because of slower solidification rates in the weld pool [25].

The measurements of the weld bead express the soundness of a weld joint. The average values obtained from a digital analysis for weld width and weld penetration depth measurements for the samples were used and are presented in Figure 8. It was observed that when laser power was increased from 800 to 1000 W, the penetration depth was reduced from 881 to 764 µm; however, it then increased to 946 µm when the laser power was 1200 W. The drop in weld penetration was due to the undercut that was observed in Figure 3b. The weld width, on the other hand, increased linearly with laser power (Figure 8a). Weld bead dimensions for three varying laser welding speeds at a constant power of 950 W are displayed in Figure 8b. The weld width decreased as heat input decreased. At high welding speeds, the molten pool becomes smaller and thus results in a reduced weld width.

The weld penetration was initially reduced from 770 to 747 µm when the welding speed was changed from 2 to 3 m/min. When weld speed was further increased to 4 m/min, the weld penetration increased to 789 µm. The reduction in weld width as welding speed increased was caused by a reduction in absorption of laser-induced plasma by the top surface of the samples at higher laser speeds [26]. The fluctuations in weld penetration depths experienced may have resulted from variances in gaps between Z225 and A6000 plates caused by inconsistencies of the clamping device [6]. One other important observation made was that in both cases (Figure 8a,b), the weld width was increased by increasing heat input. However, the increase was more pronounced when laser speed was altered. This further proves that laser welding speed is the parameter that significantly influences the weld width [4].

From the optical micrographs, it was noted that the thickness of the interfacial layer was mostly affected by a change in laser welding power compared to a change in speed. As a result, the effect of laser welding power was further studied using the SEM/EDS technique. Figure 9. Displays the microstructures of the weld metal as laser welding power was changed.

The weld microstructures for 800 and 1000 W were comparable. They revealed a fine microstructure. Laser welding is characterized by rapid cooling rates. Therefore, the atoms within steel often do not have sufficient time to diffuse into new positions. The rear-arrangement of atoms occurs quickly and creates numerous needle-shaped features displayed in Figure 9a,b [27]. As power increased to 1200 W, the grain size increased.

Figure 10 displays the SEM images showing the EDS points and the position of the EDS line scan. The IMC layer thickness increased with an increase in laser power from 800 to 1200 W (Figure 10a–c). This observation resulted from high heat input, which elevated the molten metal temperature and thus increased interaction between Fe and Al atoms, increasing the thickness of the IMC layer.

The distribution of individual elements through the weld interface was detected by EDS analysis for varying power at three different points for each sample (Figure 10a–c). The changes of the Al and Fe content are shown in Table 4. It can be observed that the composition of the phases is associated with the distance from the Z225/A6000 interface to the point of measurement. For laser powers of 800 and 1000 W, there was very little interdiffusion between Fe and Al atoms. As a result, fewer IMC phases were detected. At laser power of 1200 W, there was more interdiffusion due to increased heat input. Near Z225, the predicted phase was Fe_2_Al_5,_ and then the A6000 side contained FeAl_3_. Initially, the Al atoms spread toward steel and created a Fe_2_Al_5_ IMC layer that then nucleated within the steel/Al interface [28]. This phase is often characterized by plate-like patterns [27]. Over time, the thickness of Fe_2_Al_5_ increased and halted the fusion of solid steel and molten Al. Since the Fe atoms are larger than Al atoms, the diffusion of Fe atoms is reduced through the Fe_2_Al_5_ layer. It then created a needle-like FeAl_3_ IMC layer at the interface between the Fe_2_Al_5_ layer and Al [28]. This explains the island-shaped and needle-like structures that were detected from the optical micrographs in Figure 5b. The EDS line scan across the interfaces (Figure 10d–f) further revealed the uneven distribution of elements at the interface. At the areas where the mixing of elements occurred, the intensity of Al was much higher compared to that of Fe. This observation further explains the existence of the FeAl_3_ and Fe_2_Al_5_ phases, which are both Al-rich IMCs. The line scans also showed that the mixing of Fe and Al atoms increased as laser power increased. The XRD analyses were not able to confirm the exact IMC phases because, in most cases, the concentration of the IMC phases was below the detection limit.

### 3.2. Mechanical Properties

A microhardness test was performed to analyse the weld mechanical properties, and the results are demonstrated in Figure 11. It was detected that the hardness values of the weld metal were higher than those of Z225 and A6000. The higher hardness within the weld metal was caused by the transition of phases that occurred within the heated material, followed by rapid cooling [29]. This behavior was observed for all different combinations of laser welding parameters. The susceptibility of the weld metal to cracking was thus increased by high hardness in comparison to the base metals as reported in the work of [30]. The high hardness within the weld metal compared to the hardness of both base metals may be due to rapid cooling rates, which results in the creation of a mixture of martensite and bainite [31]. The average hardness of the weld metal was decreased with an increase in laser power (Figure 11a). When the welding speed was increased from 2 to 4 m/min, the average weld metal hardness increased from ~239.5 to ~315 HV (Figure 11b). In general, as laser welding speed increases, the grain refinement phenomenon occurs because of rapid heating and cooling rates, which lead to structural changes and consequently increased hardness levels [32]. It was observed in Figure 4 that as laser welding speed was increased, the grain size decreased. The Hall-Petch relationship tells us that through grain size refinement, the theoretical strength of the material and hardness can be achieved [32]. These results support findings observed by Sahin et al. [33] as they also noted an increase in weld metal hardness as welding speed increased. Since the IMC layer formed was too narrow, its microhardness value was not acquired. However, there were fluctuations of the weld metal hardness, which may represent different zones of the weld metal, such as the HAZ. Oyyaravelu et al. [31] obtained similar fluctuations in hardness of steel laser-welded joints. They attributed the fluctuations to the alterations in metallurgical phase components. It was further reported that the HAZ may have experienced softening, which is often caused by decarburization, which results in a carbon-denuded zone whereby relatively low thermal conductivity is observed [31]. The variations in carbon content within the weld metal were also given as a possible reason for the fluctuations [31].

The effect of laser power on the mechanical strength is displayed in Figure 12a. It was noted that tensile strength increased with an increase in laser power. This could have resulted from the higher wetting length experienced at higher laser powers, which enhanced the strength of the weld joints even though the microstructures revealed that these weld joints somewhat had greater IMCs at the interface. Reduced strength at lower laser powers was caused by a reduction in wetting length. Similar results were reported by Narsimhacharya et al. [34]. Their tensile test results showed a reduction in tensile strength as laser power was reduced. They concluded that high joint efficiency can be achieved by increasing laser power [34]. Expand on the similarities observed in your study and literature. Figure 12b presents tensile shear test results of the dissimilar weld joints at varying welding speeds. Greater failure loads were attained at lower laser welding speeds. As welding speed increased, the tensile strength was decreased. A possible reason behind this is that there was insufficient infiltration of heat across the weld joint to create a solid bond as heat input was decreased. In addition, an increase in welding speed from 2 to 4 m/min might have been significantly high for a power of 950 W and a −0.5 mm defocused beam combination as observed by Zhengwei et al. [35]. Even though the maximum strength was acquired for weld joints produced under the lowest welding speed, this procedure would not be acceptable in the automobile industry as it would result in low productivity. From the analysis of the weld grains in Figure 3b and Figure 4a, it was expected that the strength would decrease as heat input increases since the grain sizes were larger and are often characterized by a reduction in ductility [22]. As a result, there was no direct correlation between grain sizes and the strength of the weld joints. The reason for this behavior may be that the samples that were exposed to high heat input experienced grain growth, but a stronger bond was formed compared to the samples produced at lower heat inputs, whereby insufficient fusion resulted in a weak bond.

## 4. Conclusions

The findings obtained from the experiments performed on the dissimilar weld joint between Z225 and A6000 led to the following deductions: Weld bead measurements revealed that the weld width increased with an increase in heat input, whereas weld penetration fluctuated. Laser welding speed was the parameter that affected the weld width the most. Increasing heat input increased the presence of island-shaped structures and needle-like structures, which were identified as Fe_2_Al_5_ and FeAl_3,_ respectively. The IMC layer was increased with an increase in laser power. An increase in heat input improved the tensile strength of the joints. The microhardness of the weld bead was higher than the hardness of both base metals. Laser power of 1200 W and 3 m/min laser speed were identified as ideal parameters and produced a weld joint of the highest strength (1.79 kN).

## Figures and Tables

**Figure 1 materials-15-00543-f001:**
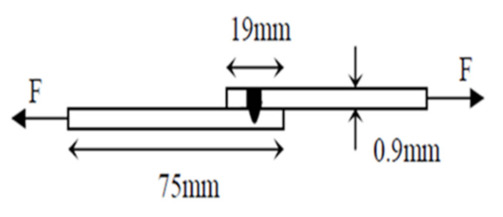
Specimen used for tensile shear strength test indicating direction of applied stress [16].

**Figure 2 materials-15-00543-f002:**
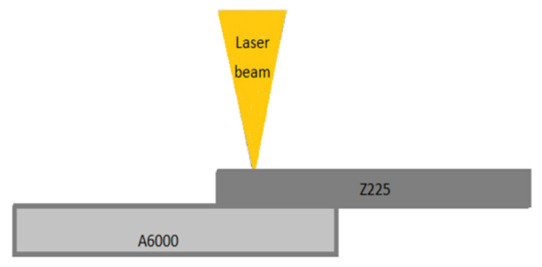
Schematic representation of laser welding process of a lap joint [4].

**Figure 3 materials-15-00543-f003:**
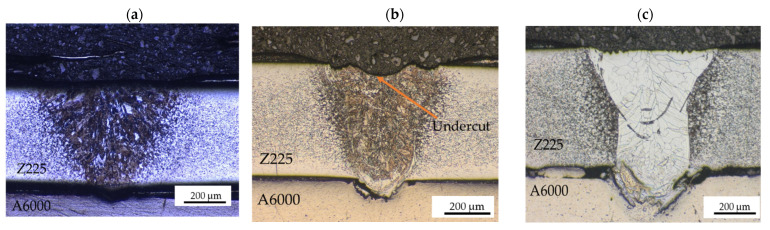
Optical micrographs for 3 m/min and different laser powers at 20× showing: (**a**) insufficient fusion at 800 W; (**b**) undercutting at 1000 W; (**c**) columnar grain structure at 1200 W.

**Figure 4 materials-15-00543-f004:**
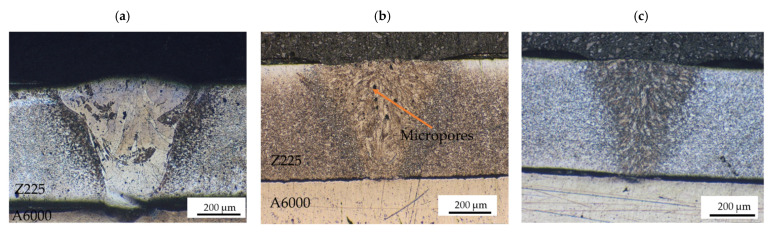
Optical micrographs for 950 W and different laser welding speeds at 20× magnification showing (**a**) columnar grain structure at 2 m/min; (**b**) micropores at 3 m/min; (**c**) insufficient fusion at 4 m/min.

**Figure 5 materials-15-00543-f005:**
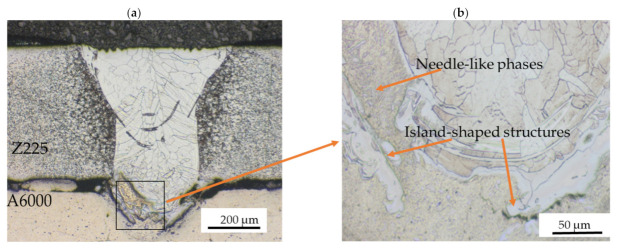
Optical micrographs for 1200 W and 3 m/min showing (**a**) columnar grain structure at 20×; (**b**) weld interface with irregular phases at 50×.

**Figure 6 materials-15-00543-f006:**
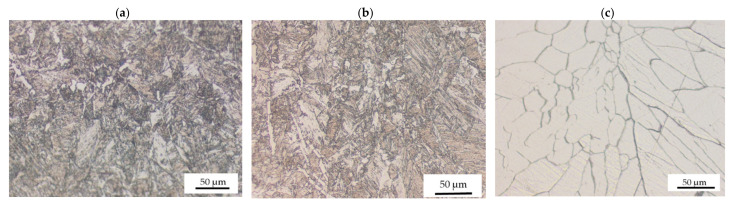
Optical micrographs for 3 m/min and different laser powers at 50× showing: (**a**) fine grain structure 800 W; (**b**) fine grain structure at 1000 W; (**c**) large columnar grain structure at 1200 W.

**Figure 7 materials-15-00543-f007:**
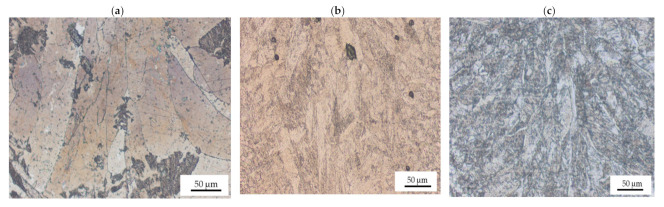
Optical micrographs for 950 W and different laser welding speeds at 50× magnification showing (**a**) large columnar grain structure at 2 m/min; (**b**) micropores at 3 m/min; (**c**) fine equiaxed grain structure at 4 m/min.

**Figure 8 materials-15-00543-f008:**
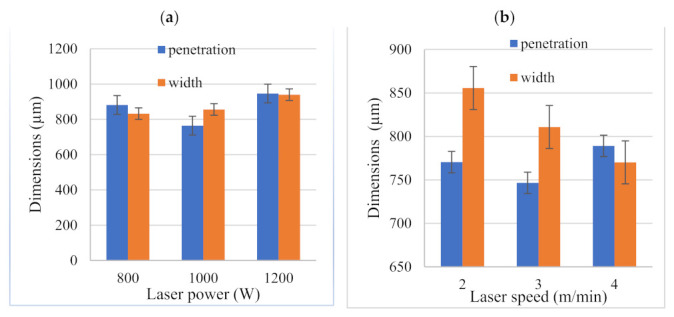
Effect of system parameters on the weld bead dimensions, (**a**) effect of laser power; (**b**) effect of laser welding speed.

**Figure 9 materials-15-00543-f009:**
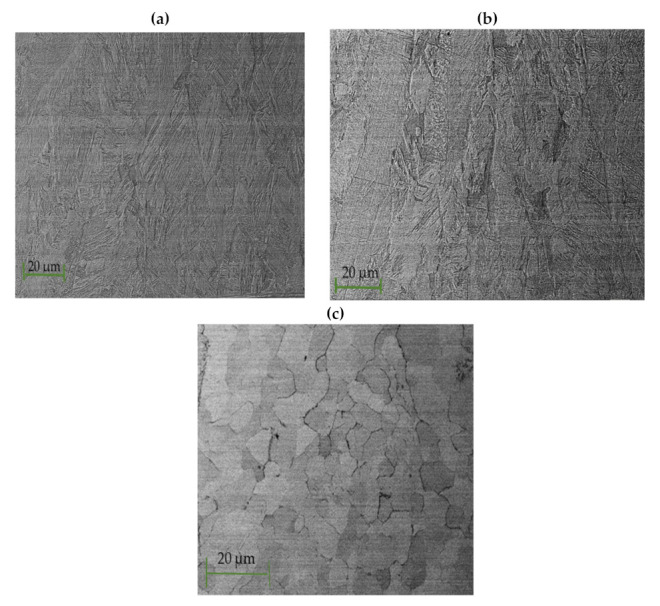
SEM images showing the effect of laser welding power on the microstructure of the weld metal, (**a**) 800 W; (**b**) 1000 W; (**c**) 1200 W.

**Figure 10 materials-15-00543-f010:**
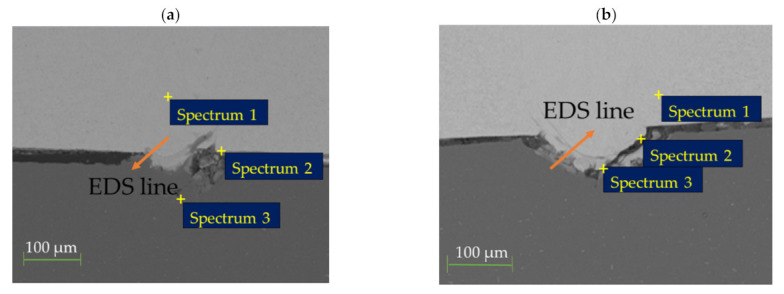
SEM images showing direction of EDS line scan and EDS points; (**a**,**d**) 800 W; (**b**,**e**) 1000 W; (**c**,**f**) 1200 W.

**Figure 11 materials-15-00543-f011:**
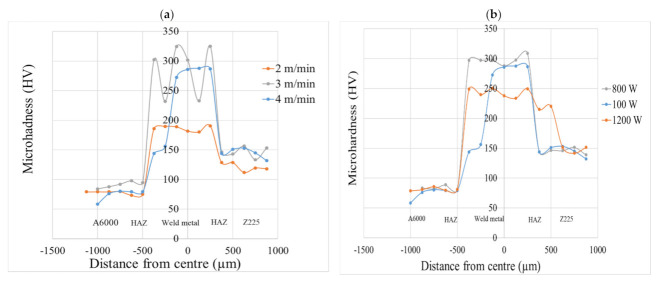
Effect of laser parameters on microhardness; (**a**) effect of laser power; (**b**) effect of laser welding speed.

**Figure 12 materials-15-00543-f012:**
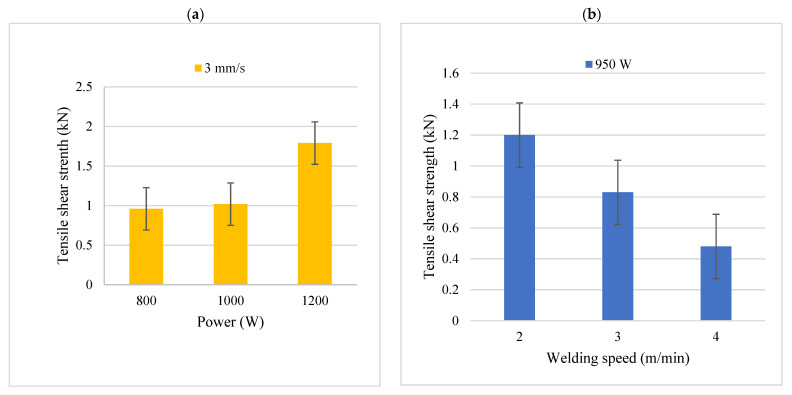
Effect of laser parameters on tensile shear strength; (**a**) effect of laser power; (**b**) effect of laser welding speed.

**Table 1 materials-15-00543-t001:** Chemical composition of parent metals [15].

Elements (wt.%)			
	Si	Mg	Cu
A6000	1.0	0.5	<0.1
Material	C	Si	P
Z225	<0.15	<0.60	<0.05

**Table 2 materials-15-00543-t002:** Mechanical properties of parent metals.

Parent Metal	Yield Strength (MPa)	Ultimate Tensile Strength (MPa)	Elongation (%)
A6000	225	303	15
Z225	350	375	35

**Table 3 materials-15-00543-t003:** Welding system parameters.

System Parameters		
Power (W)	Welding Speed (m/min)	Heat Input (10^5^ J/mm)
800	3	16
950	2	28.5
950	3	19
950	4	14.25
1000	3	20
1200	3	24

**Table 4 materials-15-00543-t004:** EDS elemental distribution and probable IMC phases.

Power	800 W			1000 W			1200 W		
Weight (%)	Fe	Al	Phase	Fe	Al	Phase	Fe	Al	Phase
Spectrum 1	100.0	0	Fe-rich	100	0	Fe-rich	77.60	22.40	Fe-rich
Spectrum 2	94.05	5.95	Fe-rich	24.04	75.96	FeAl_3_	34.13	65.87	Fe_2_Al_5_
Spectrum 3	0.24	99.76	Al-rich	1.08	98.92	Al-rich	20.21	79.79	Al-rich

## Data Availability

Not applicable.

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
