# Peer review of "Microstructural Characterization and Mechanical Properties of Laser Beam-Welded Dissimilar Joints between A6000 Aluminum Alloy and Galvanized Steel"

_materials, 2022, doi:10.3390/ma15020543_

Round 1
Reviewer 1 Report
The manuscript entitled "Microstructural characterization and mechanical properties of laser beam welded dissimilar joints between A6000 aluminium alloy and galvanized steel" has been investigated in detail. The topic addressed in the manuscript is potentially interesting and the manuscript contains some practical meanings, however, there are a few issues which should be addressed by the authors:
In the introduction, the motivation and contribution should be described clearly.
Figure 6 shows that weld width grows with laser power, whereas penetration depth initially decreases and later increases. This pattern of laser power influence on penetration depth should be examined with various explanations.
Similarly, the welding speed influence on weld width is well predicted, but the pattern for penetration depth deviates from the expected trend. Why does penetration depth increase with welding speed?
The parametric influence on dilution rate, and hence on weld strength and microstructure, may be highlighted.
The section "Results and Discussion" could be modified to be more highlighting and argumentative. The author should do a more in-depth investigation of why the tested outcomes were obtained.
Overall, the work is well-written and organized. The observations are precise and informative. Only in terms of depth of discussion does the paper fall short. The authors should work on this and then submit a modified manuscript. While submitting the revised paper highlight the changes with colored text.
Author Response
Authors notes to reviewer 1
Dear Reviewer.
Please note the changes you recommended for the article
- The motivation and contribution of the article were added at the end of the introduction section.
- Reasons behind the fluctuations in weld bead dimensions were provided.
- The influence of parameters on the weld strength and microstructure were highlighted
- Literature was added to the results and discussion section to enhance the arguments.
- Spelling and grammar errors were corrected
Thank you for the input and looking forward to hearing further comments
Reviewer 2 Report
The authors performed dissimilar laser welding of galvanized steel and A6000 aluminum alloy sheets. The topic is meaningful, and detailed experimental setup and processes were stated. Yet, this paper is more like a technical report rather than an academic article. Here are some tips that the authors can consider.
- In Table 1, the word ‘Materials’ could be removed. Otherwise it is kind of misleading.
- In Table 3, the magnitude of heat input is incorrect. The unit of heat input should be 105 J/mm if I didn’t calculate them wrong.
- The microstructure characterization is insufficient. Since the authors have employed SEM to check the microstructure of the weld seam, it would be better to take some images of the microstructure with higher magnification. Aided by SEM and EDS, more details of the microstructure could be obtained.
- In Fig. 7, I encourage the authors to add the line scan data to illustrate the element distribution features.
- In Fig. 8, the digits on the x-axis are missing. Please add them since it’s the distribution of microhardness as a function of position.
- As the authors described, the Zn coating should have played an important role in the process of laser welding. However, the effects of Zn in this paper were not analyzed well. All people specialized in welding have known that Zn may lead to pores in the weld seams, but how did the Zn coating influence the present case?
Author Response
Authors notes to Reviewer 2
Dear Reviewer.
The following changes were made as per your request to enhance the article
- The word ‘‘materials’’ in Table 1 was removed.
- The magnitude of heat input in Table 3 was adjusted.
- More optical micrographs at higher magnification were added to study the microstructure of the weld. SEM images and EDS line scan were also included to clearly show the interaction of elements at the interface and how they reacted to form IMC phases.
- The x-axis digits (distance in microns) for microhardness curves were added.
- The effect of zinc coating on the project was also highlighted.
- More literature was added under results and discussion to enhance the arguments
Looking forward to hearing from you. Thank you so much for the inputs. The authors hope that the changes will meet your expectations.
Reviewer 3 Report
- Please address the purpose of this study at the end of the introduction section. What is the scientific contribution of this manuscript?
- Page 4, please add a schematic to show the line measurements of the hardness.
- Page 4, you wrote “A Zwick/Roell electro-chemical machines”, please check if it is electro-chemical or electromechanical or something else.
- Page 5, please change the subtitle of section 3 from “3. Resutls” to “3. Results and Discussion”.
- Figure 3(a), what are black dots (pores?) in the aluminum alloy base metal, due to polishing or not? If so, please re-ground and re-polish this sample to take a better optical image. If not, please explain the formation of the black dots.
- Page 8, “However, there were fluctuations of the weld metal hardness which may represent different zones of the weld metal.” Please give more clear explanation for the fluctuated hardness in the weld bead (fig. 8b). Again, please clarify how the hardness line measurements across the sample in the experimental section.
Author Response
Authors notes to reviewer 3
Dear reviewer
The following modifications were made as per your request to enhance the article.
- The motivation, purpose and contribution of the article were added at the end of the introduction section.
- Figure shows the areas where the microhardness indentations were performed.
- The name of the Zwick/Roell machine used for the tensile tests was changed
- The subtitle “3. Results” was changed to “3. Results and discussion”.
- The micrograph image for Figure 3(a) was repolished to remove the dots and a new image was used.
- Literature was consulted to address the fluctuations observed in microhardness values within the weld.
Thank you so much for the input and looking forward to hearing from you.
Round 2
Reviewer 1 Report
The authors have modified their manuscript in line with the reviewer's suggestions. The manuscript can now be accepted for publication.
Reviewer 2 Report
The authors have revised the manuscript according to my comments. I judge that this paper has reached the standard of Meterials, which could be accepted.